# Differences in Immunological Evasion of the Delta (B.1.617.2) and Omicron (B.1.1.529) SARS-CoV-2 Variants: A Retrospective Study on the Veneto Region’s Population

**DOI:** 10.3390/ijerph19138179

**Published:** 2022-07-04

**Authors:** Silvia Cocchio, Federico Zabeo, Giacomo Facchin, Nicolò Piva, Giovanni Venturato, Thomas Marcon, Mario Saia, Michele Tonon, Michele Mongillo, Filippo Da Re, Francesca Russo, Vincenzo Baldo

**Affiliations:** 1Department of Cardiac Thoracic and Vascular Sciences and Public Health, University of Padua, 35131 Padua, Italy; silvia.cocchio@unipd.it (S.C.); federico.zabeo@unipd.it (F.Z.); giacomo.facchin@studenti.unipd.it (G.F.); nicolo.piva@studenti.unipd.it (N.P.); giovanni.venturato.1@studenti.unipd.it (G.V.); 2Azienda Zero of Veneto Region, 35100 Padua, Italy; thomas.marcon@azero.veneto.it (T.M.); mario.saia@azero.veneto.it (M.S.); 3Regional Directorate of Prevention, Food Safety, Veterinary, Public Health—Veneto Region, 30123 Venice, Italy; michele.tonon@regione.veneto.it (M.T.); michele.mongillo@regione.veneto.it (M.M.); filippo.dare@regione.veneto.it (F.D.R.); francesca.russo@regione.veneto.it (F.R.)

**Keywords:** SARS-CoV-2, VOCs, Omicron, Delta, vaccines, reinfection, immunological evasion, Survival Analysis, Veneto region

## Abstract

In December 2021–January 2022 the Veneto region in Italy faced an unprecedented wave of SARS-CoV-2 infections, even though both the vaccine coverage and the number of previously infected individuals keep increasing. In this study we address the protection against the SARS-CoV-2 infection offered by natural immunity and a three-dose regimen through a retrospective study based on Veneto’s regional databases. In particular, we compared these protection levels during two distinct periods respectively representative of the Delta (B.1.617.2) and the Omicron (B.1.1.529) variants, in order to investigate and quantify the immunological evasion, especially of the Omicron. For each period we compared the incidence rate of infection among the population with various immunological protections against SARS-CoV-2 and performed a multivariable proportional hazard Cox binomial regression to assess the effectiveness afforded by both forms of active immunization. We found out that a previous SARS-CoV-2 infection (irrespective of its timing) offers 85% (83–87%) and 36% (33–39%) protection against being reinfected by Delta and Omicron, respectively. In addition, we estimated the third dose to be more effective in both periods and to have a minor proportional loss of effectiveness due to the rise of the Omicron variant, with an afforded effectiveness against SARS-CoV-2 Delta and Omicron infection of 97% (96–97%) and 47% (45–48%), respectively. Our findings suggest that viral variant factors may affect any form of active immunization but that receiving a booster vaccination cycle is more effective and less variable than natural immunity in terms of afforded protection against SARS-CoV-2 infections.

## 1. Introduction

As of 14 March 2022, the COVID-19 pandemic continues to have a huge impact on the public health and economy of many countries. To date, it is estimated that SARS-CoV-2 has caused more than 450,000,000 cases and approximately 6,000,000 deaths worldwide [1]. 

The Veneto region, which is located in the northeastern part of Italy and has a population of about 5 million inhabitants [2], also experienced an unprecedented health emergency, recording 1,380,410 SARS-CoV-2 confirmed cases and 13,965 deaths due to COVID-19 [3].

Several public health interventions were introduced in the Veneto region to counteract the COVID-19 emergency, such as lockdowns, school closures, and social distancing. Moreover, on 27 December 2020 Italy started rolling out an anti-COVID-19 vaccination campaign using EMA-approved vaccines: BNT162b2 (Comirnaty, Pfizer–BioNTech, Mainz, Germany) was the most widely used vaccine, followed by mRNA-1273 (Spikevax, Moderna Biotech, Madrid, Spain), ChAdOx1 (Vaxzevria, AstraZeneca, Södertälje, Sweden), and Ad26.COV2.S (Janssen, Janssen-Cilag International NV, Beerse, Belgium) [4]. The vaccination campaign was initially open to all adults (over 18 years old) prioritizing vulnerable individuals (those 80+ years of age, individuals with pre-existing comorbidities, and those living in nursing homes) and health care workers. From 3 June 2021 vaccinations became available for all individuals over 12 years old, and finally, from 15 December 2021, the vaccination campaign was also extended to children between 5 and 11 years old with a reduced dosage (one third of the regular dosage) [5].

So far, more than 10 million doses have been administered in the Veneto region, with 85.6% of the population eligible who have received at least one dose and 84.6% fully vaccinated individuals. [6].

Concerns over the waning immunity of anti-COVID-19 vaccinations, which have been motivated by growing scientific evidence [7], called for the administration of a third “booster” dose to restore protection against infections and their complications to its previous level [8].

Italy started a “supplementary” dose campaign while trying to further improve primary-cycle coverage. In particular, the Italian Ministry of Health provided preliminary guidance on the administration of “additional doses” as part of a primary cycle for some categories of immunocompromised individuals and “booster doses” for all other eligible subjects who have been fully vaccinated for at least 6 months, subsequently lowered to 4 months [9]. Italian regions started administering additional doses on 20 September 2021 using either mRNA BNT162b2 or mRNA-1237 regardless of the vaccine previously used. Thereafter, on 27 September 2021, Italian regions also started administering booster doses [10] using mRNA BNT162b2 and mRNA-1237 (50 μg). Third doses are currently available only for individuals over 12 years old.

To date, more than 3 million third doses have been administered by the Veneto region [6], including booster doses for subjects who received the Ad26.COV2.S single-dose vaccine. At the time of writing, about 65% of the population is estimated to have received a third-dose vaccine [11].

Despite the continuous rise in the coverage of both primary cycles and third doses, the Veneto region recently faced a singular wave of infection with more than 480,000 new cases occurring just in January 2022 [12]. Different causes were considered to explain this surge, including the seasonal behavior of the virus, relaxation of public health interventions, and the spreading of the new Omicron (B.1.1.529) variant which gradually wiped out the previous Delta (B.1.617.2) variant. The Omicron variant, in fact, turned out to be characterized by a higher transmission rate [13] and several mutations in the spike protein [14] leading to a reduction in the protection afforded by both vaccine-induced [15,16,17] and natural immunity (a growing trend for the relative risk of reinfection from 6 December 2021 was revealed [18]).

In fact, the World Health Organization (WHO) included Omicron in its list of Variants of Concern (VOCs) [19].

Fortunately, evidence from the scientific community showed that the above-mentioned increase in transmissibility did not generally coincide with an increase in the severity and fatality of the disease, especially among vaccinated subjects [20,21].

Even if further evidence may be required to better understand the contribution of each of the above-mentioned factors, together with other possible elements, Omicron’s immunological evasion might play a role in determining the future trajectory of the epidemic; indeed, it could determine a new chapter in the COVID-19 pandemic [22] increasing the likelihood of SARS-CoV-2 to become endemic. Unsurprisingly, there is currently international interest in defining the impact of infection and reinfection among vaccinated subjects, since many studies are showing that the circulation of SARS-CoV-2 might continue even after reaching herd immunity through natural infection or vaccination [23].

For all these reasons, our work has been designed to evaluate the effectiveness of both natural immunity and the three-dose regimen against Delta and Omicron infections.

## 2. Materials and Methods

### 2.1. Data Source

The source of the data used in the present study has already been extensively discussed [24]. Briefly, anonymized data was retrieved from regional databases compiled through a mandatory reporting system. The present analysis entirely relies on databases containing general information about all the RT-PCR SARS-CoV-2 tests and the anti-COVID-19 vaccinations performed in the Veneto region together with general demographic information (sex and age) about the individuals being tested or vaccinated. Rapid antigen tests were deliberately not included in our analysis. The latest available update was on 25 January 2022 and included both databases.

We considered individuals as infected when they tested positive to at least one RT-PCR test. To exclude errors induced by prolonged viral shedding, we considered subjects as reinfected when they tested positive after more than 90 days from a preceding positive test, which is a definition consistent with the one commonly used for epidemiological purposes [25].

Before performing analysis, all data were cleaned to exclude probable errors during the registration of vaccination data: in particular, only subjects whose vaccination schedule followed the rules and timing approved by the Italian Ministry of Health and for whom vaccine typology was specified and matched with EMA-approved vaccines were included.

To assess the waning of the protection afforded by prior infection or by a complete anti-COVID-19 vaccination cycle (including third dose) with the rise of the new Omicron variant, we performed and compared analyses on two distinct, equal-length periods. We defined the “Delta period” as the period between 1 and 25 November 2021 and the “Omicron period” as the one between 1 and 25 January 2022. The Omicron period was characterized by a predominance of the Omicron variant, which reached 66.1% of all SARS-CoV-2 confirmed cases in the Veneto region on 3 January 2022 and kept spreading rapidly, whereas in the other period almost all SARS-CoV-2 infections were due to the Delta variant, since the percentage of other variants was negligible [26].

For both periods, we retrospectively defined cohort populations according to appropriate eligibility criteria.

### 2.2. Study Design

For each study period we retrospectively defined three distinct populations. We firstly defined the “reference group” as consisting of individuals tested at least once during the period under consideration who had never tested positive for SARS-CoV-2 nor received any vaccine dose before the beginning of the study period. Secondly, we defined the “natural-immunity group” as the subgroup of subjects who received at least one RT-PCR during the study period, had already tested positive to SARS-CoV-2 more than 90 days before, and had not received any vaccine dose by the beginning of the study period. As stated previously, to avoid misclassification for “reinfection” and considering the protection afforded by multiple preceding infections, we excluded from the natural immunity population those subjects who tested positive within 90 days preceding the beginning of the study and those who had already experienced a reinfection before the same date. Finally, we defined the “three-dose regimen group” as including all subjects who were tested at least once during the study period, received the third dose of anti-COVID-19 vaccination before, and did not test positive by the beginning of the study period. Here the term “three-dose regimen group” has been chosen because this group might include both subjects who received an additional or a booster dose, since our data source does not provide enough information to differentiate them.

For each subject we computed his/her follow-up as the number of days elapsed between the starting date of the study period and the date of infection/reinfection or the date of first vaccination (if they were unvaccinated when enrolled in the study) or the day after the end of the study period, whichever came first. Thus, we performed a multivariable Cox binomial regression considering the presence of an infection as the dependent event and sex, age at the first available sampling date, number of tests performed in the considered period, possible presence (and timing) of a preceding infection, possible presence of a three-dose vaccination cycle, and number of days of follow-up (FU) as dependent covariates. 

We categorized the timing of a preceding infection as follows: recent infections (3 to 6 months before the baseline), medium-distance infections (6 to 9 months before the baseline), medium-/long-distance infections (9 to 12 months before the baseline), and long-distance infections (more than 12 months before the baseline).

We did not further adjust by the timing of the third-dose vaccination because, considering that third doses were available from the end of September 2021 and the enrolment in our study ended on 1 January 2022, there are no subjects in the cohort that received the third dose more than 120 days before enrolling.

The protection afforded by a preceding infection (with its timing) and the effectiveness of the three-dose regimen were then estimated as 1-adjOR, where adjOR stands for the adjusted odds ratio between the risk of the event between the natural-immunity or the three-dose regimen group and the reference one.

The same method was used to estimate the global protection offered by a preceding infection (irrespective of its timing), considering all subjects with a preceding infection as a single category.

### 2.3. Statistical Analysis

We summarized the data using percentages, positivity rates (i.e., the number of distinct subjects who tested positive over the number of tests performed), incidence rates (i.e., the number of distinct subjects who tested positive over the number of person-days of follow-up of the considered cohort), and mean values with their 95% confidence intervals or median with their IQR, as appropriate.

Comparison of the survival times among different subgroups are presented through the cumulative hazard function, which represents the failure rate within the considered population to varying of the number of days elapsed from the baseline.

To describe the effect of specific covariates on survival we adopted multivariable Cox proportional hazards regressions. A *p*-value below 5% was considered statistically significant.

Data linkage, cleaning, and all statistical analyses were performed with specific libraries of Python (Python 3.7.0., Guido van Rossum, CWI, Amsterdam, The Netherlands).

## 3. Results

### 3.1. Delta Period

After applying the eligibility criteria described above, the study population consisted of 102,676 subjects. The median number of tests performed by each of them is 1 (IQR 1-2), and the overall number of tests performed among the study population is 154,178.

Among the whole cohort, 78.1% (N. 80,238) subjects belong to the reference group, 16.4% (N. 16,886) belong to the three-dose regimen group, and the remaining ones belong to the natural immunity group.

Among the 5,552 previously infected subjects, 54.3% (N. 3,014) tested positive to SARS-CoV-2 from 9 to 12 months before the beginning of the study period, 25.9% (N. 1,438) from 6 to 9 months before, and 10.8% (N. 600) from 3 to 6 months before; the remaining 500 subjects tested positive more than 12 months before the Delta period began.

The number of subjects who got an infection in the considered study period is 15,483, which accounts for 15.1% of the whole study population. Among all these infections, 97.7% (N. 15,133) occurred in the reference group and 170 in the three-dose regimen group, whereas 180 were reinfections. Both having a preceding infection and receiving a third dose significantly decreases the risk of contracting the virus during the period under consideration (Figure 1).

A multivariable Cox model estimates at 85% (83–87%) the protection against SARS-CoV-2 infection given by natural immunity (irrespective of the timing of the previous infection) and at 97% (96–97%) the effectiveness afforded by the three-dose regimen in preventing SARS-CoV-2 infections.

The distribution of infections, the positivity rate, the incidence rate, and the adjusted odds in the risk of contracting the virus as stratified per different subgroups based on the presence (and possibly the timing) of a prior immunization are summarized in Table 1.

### 3.2. Omicron Period

After applying the eligibility criteria described above, the study population consists of 116,022 subjects. The median number of tests performed by each of them is 1 (IQR 1-2), and the overall number of tests performed among the study population is 171,943.

Among the whole cohort, 58.4% (N. 67,721) subjects belong to the reference group, 36.7% (N. 42,530) belong to the three-dose regimen group, and the remaining ones composed the natural-immunity group.

Among the 5,771 previously infected subjects, 49.5% (N. 2,857) tested positive to SARS-CoV-2 more than 12 months before the beginning of the study period, 24.7% (N. 1,424) from 9 to 12 months before, and 18% (N. 1,043) from 3 to 6 months before; the remaining 447 subjects tested positive from 6 to 9 months before the Omicron period began.

The number of subjects who got an infection in the considered study period is 41,146, which accounts for 35.5% of the whole study population. Among all these infections, 69.8% (N. 28,715) occurred in the reference group and 26% (N. 10,690) in the three-dose regimen group, whereas 1,741 were reinfections. Both having a preceding infection and receiving a third dose slightly decreases the risk of contracting the virus during the considered period (Figure 2).

A multivariable Cox model estimates at 36% (33–39%) the protection against SARS-CoV-2 infection given by natural immunity (irrespective of the timing of the previous infection) and at 47% (45–48%) the effectiveness afforded by a three-dose regimen in preventing SARS-CoV-2 infections.

The distribution of infection, the positivity rate, the incidence rate, and the adjusted odds ratio in the risk of contracting the virus as stratified per different subgroups based on the presence (and possibly the timing) of a prior immunization are summarized in Table 2.

## 4. Discussion

Our findings, consistently with recent epidemiological trends [27], confirm that the arrival of the Omicron variant coincided with an increase in the incidence of SARS-CoV-2 cases which in our study population approximatively tripled from the Delta period to the Omicron period.

We showed that in both the periods SARS-CoV-2–naïve subjects are more at risk than those with previous natural or vaccine-induced immunity, confirming the significant neutralizing effect afforded by any type of previous antigenic exposure [28,29].

We observe that subjects who suffer the highest proportional increase in the incidence rate of infection during the Omicron period are those who received their third dose (+2,805%) and previously infected individuals (+1,084%), whereas SARS-CoV-2–naïve individuals registered the least marked rise (+175%). This finding might suggest that the most relevant impact of the Omicron variant is due to its increased immune evasion rather than to an increase in its transmissibility.

For what concerns the Delta period, we found that the overall risk of being infected at the end of the period was 1% for subjects belonging to the booster group, lower than 4% for those from the natural immunity group, and 20% among SARS-CoV-2–naïve individuals.

This finding was further confirmed by running a multivariable Cox model, which estimated the effectiveness of a recent third dose against SARS-CoV-2 Delta infection at 97% (96–97%) whereas it evaluated the protection afforded by a previous infection to be equal to 85% (83–87%); this suggests that even if natural immunity significantly protects subjects against the SARS-CoV-2 Delta infection, having recently received a booster dose is still more effective.

On the other hand, the estimated risks of contracting SARS-CoV-2 infection during the Omicron period were 40% for the reference group, 30% for the natural immunity group, and 25% for the group that received third doses. According to the Cox model, the adjusted effectiveness of a recent third dose against contracting the SARS-CoV-2 Omicron infection is 47% (45–48%) whereas the protection afforded by a previous infection is 36% (33–39%).

This evidence unfortunately suggests a diminished protection from Omicron when compared to Delta in both natural (−57.6%)—consistently with other studies [30]—and vaccine-induced immunity (−51.5%). Interestingly, the most remarkable proportional decrease in the protection afforded seems to involve natural immunity, which also turns out to be again less effective than a fresh third dose in preventing SARS-CoV-2 infection.

Further stratifying the natural immunity group on the timing of the first infection, we found that during the Delta period the least protected [76% (66–83%)] from reinfections were those who had their first infection more than 12 months before. This latter protection, particularly, turns out to be significantly lower than the one achieved with an infection occurring between 9 and 12 months before the starting date [87% (84–90%)].

More recent prior infections are also significantly protective [85% (80–89%) for infections occurring between 6 and 9 months before the baseline, 84% (74–91%) for those occurring between 3 and 6 months before]. However, probably because of the low number of subjects with such prior infections in our cohort, their protection does not result significantly different from the ones afforded by prior infections with a different timing. These findings suggest that during the Delta scenario natural immunity slightly (−10% approximately) wanes over time, especially when more than 12 months have elapsed since the previous infection.

The same analysis conducted for the Omicron period showed no statistically significant differences, suggesting that the timing of the previous exposure seems to relatively lose importance.

It is important to mention that the case definition we adopted for reinfection is just one of the possible case definitions proposed by public health entities and that this choice unavoidably influences the results. For instance, with the technical report of 8 April 2021, ECDC proposed to consider a subject reinfected when he/she tested positive after more than 60 days from a preceding positive test. Since the aim of our study is to compare protection levels in different periods rather than to describe the burden of reinfection as a whole, we decided to use the “>90 days threshold” in order to be more conservative and not overestimate the number of reinfections.

This study has several limitations, including the retrospective methodology, and our findings should be interpreted carefully. To begin with, since third doses started being administered at the end of September 2021, our results are relative to only a short period of observation for the three-dose regimen group. This implies that our results about the effectiveness of a three-dose regimen are restricted to subjects who received their third dose recently (i.e., within 120 days) and do not provide any information about how Delta and Omicron variants escape immunity against infections provided by a third dose received earlier. Moreover, some factors could have biased the protection afforded by natural and vaccine-induced immunity. Firstly, since genomic sequencing was not available in our data we assumed variants by proxy, i.e., assuming that each confirmed SARS-CoV-2 case in the Delta/Omicron period was due to the homonymous variant. However, although dominant, Omicron cases were not present alone during what has been described as the Omicron period. This is a particularly important aspect, in that the copresence of both Delta and Omicron variants during what we called the Omicron period could have led to an overestimation of the protection levels against the Omicron variant due to a dilution effect. In addition, between November and December 2021 the number of rapid swabs performed rapidly increased, with antigenic swabs replacing molecular ones in principal screening activities [31]; moreover, from the end of December 2021 they gained the same diagnostic value as RT-PCR tests, cancelling the need to perform a molecular swab to confirm a positive antigenic one [32]: due to this inconstant and nonhomogeneous utilization of antigenic rapid swabs and their low level of sensitivity, particularly in asymptomatic subjects [33], we preferred not include them into our analysis. We want also to stress that our analysis relies entirely on regional databases compiled through a mandatory reporting system and contains very general information: this means that, for our analysis, we were obliged to consider a non-exhaustive list of potential confounding factors. Moreover, the data at our disposal do not provide any information about non-COVID-19 related deaths or relocations, implying that we might have overestimated the follow-up of some subjects for which a right censoring should have been performed instead. Finally, it has to be mentioned that our results could be affected by underascertainment of SARS-CoV-2 cases, especially where previous infections are concerned. We believe that it is quite impossible to infer how undocumented infections affected our findings, since it is difficult to infer whether they distributed homogeneously or non-homogeneously in the subpopulations and in the time periods considered in our analysis. However, it is plausible that underascertainment of SARS-CoV-2 cases increases the uncertainty in the estimation we provided for the protection levels of both natural and vaccine-induced immunity. Thus, we believe it is necessary to mention this public health issue too as a limitation of our study.

Nevertheless, we managed to conduct statistically significant analysis on large volumes of regional data.

In our opinion, further studies are needed to evaluate how protection against SARS-CoV-2 infection wanes in a longer period and possibly with the rise of new variants that may spread, such as the BA.2 Omicron subvariant that has recently started spreading rapidly. Also, the limited time intervals we considered for the present study did not allow us to perform a robust analysis on the duration and protection level against infection afforded by “hybrid immunity” (i.e., immunity achieved by infection combined with vaccination [34]), and we believe that these are arguments which should be deepened.

In addition, more stratified comparisons—for example, those differentiating by the type of vaccines received—should be done. Finally, we think that all these analyses should be extended to other outcomes such as hospitalization, intensive-care therapy admission, and deaths.

## 5. Conclusions

In conclusion, our study highlights that having a previous infection and receiving a third-dose vaccination provide extremely high protection against the SARS-CoV-2 Delta infection, whereas the new Omicron variant seems significantly to evade their neutralization effect. In particular, it turns out that receiving a three-dose vaccination cycle gives higher protection against both VOCs, and our findings also suggest that naturally immunized subjects are those who experience the most significant proportional decrease in their protection with the rise of the Omicron variant. This evidence indicates that, even if viral variant factors could affect any form of active immunization, receiving a three-dose vaccination cycle is more effective and less variable than contracting an infection in terms of afforded protection against SARS-CoV-2 infections.

## Figures and Tables

**Figure 1 ijerph-19-08179-f001:**
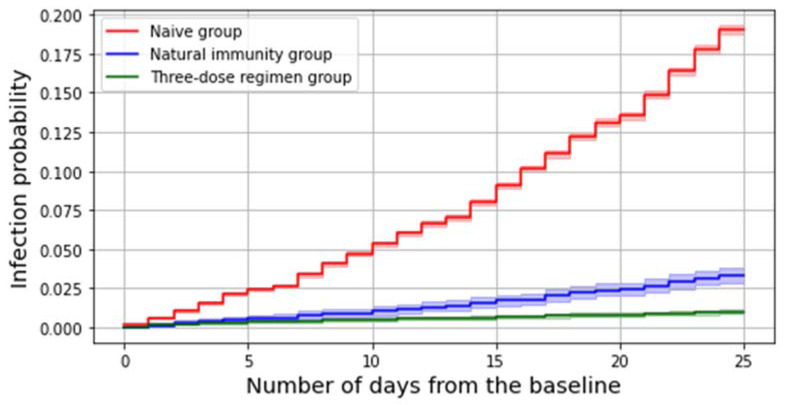
Failure curves representing the cumulative probability of being infected during the Delta period among different groups.

**Figure 2 ijerph-19-08179-f002:**
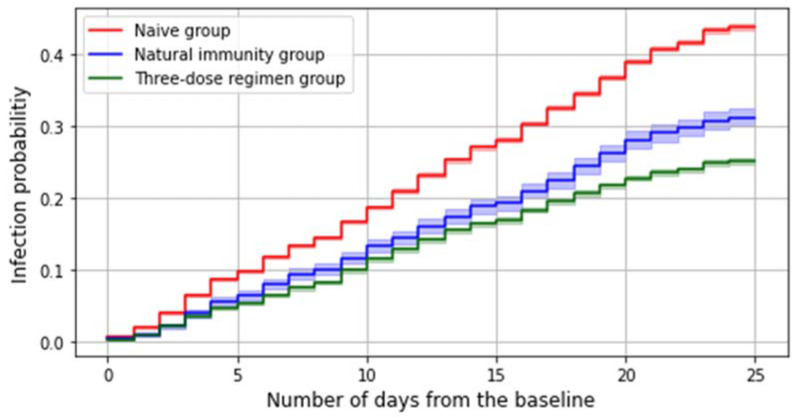
Failure curve representing the cumulative probability of having a first infection during the Omicron period.

**Table 1 ijerph-19-08179-t001:** Positivity rate and incidence of infection, by immunity status (Delta period).

Immunity Status	Subjects (*n*)	Swabs (*n*)	Person-Days (*n*)	Infections (*n*)	Positivity Rate (%)	Incidence Rate (×10,000 Person-Days)	adjOR (95% C.I.)
No previous infection	80,238	11,381	1,836,179	15,133	13.3	82.4	Ref.
Infected 3+ months before baseline	5,552	8,461	133,972	190	2.2	14.2	0.15 (0.13–0.17)
Infected 12+ months before baseline	600	947	14,509	33	3.5	22.7	0.24 (0.17–0.34)
Infected 9 to 12 months before baseline	3,014	4,789	72,886	87	1.8	11.9	0.13 (0.10–0.16)
Infected 6 to 9 months before baseline	1,438	2,042	34,843	45	2.2	12.9	0.15 (0.11–0.20)
Infected 3 to 6 months before baseline	500	683	11,734	15	2.2	12.8	0.16 (0.09–0.26)
Third dose within 120 days from the baseline	16,886	31,911	419,765	170	0.5	4	0.03 (0.03–0.04)
Total	102,676	15,418	2,389,916	15,483	10	64.8	−

**Table 2 ijerph-19-08179-t002:** Positivity rate and incidence of infection, by immunity status (Omicron period).

Immunity Status	Subjects (*n*)	Swabs (*n*)	Person-Days (*n*)	Infections (*n*)	Positivity Rate (%)	Incidence Rate (×10,000 Person-Days)	adjOR (95% C.I.)
No previous infection	67,721	90,819	1,266,730	28,715	31.6	226.7	Ref.
Infected 3+ months before baseline	5,771	8,694	117,473	1,741	20	148.2	0.64 (0.61–0.67)
Infected 12+ months before baseline	2,857	4,757	58,686	873	18.4	148.8	0.64 (0.59–0.68)
Infected 9 to 12 months before baseline	1,424	1,984	28,742	436	22	151.7	0.67 (0.61–0.73)
Infected 6 to 9 months before baseline	447	599	9,311	132	22	141.8	0.62 (0.52–0.74)
Infected 3 to 6 months before baseline	1,043	1,354	20,734	300	22.2	144.7	0.64 (0.57–0.71)
Third dose within 120 days from the baseline	4,253	7,243	919,823	1,069	14.8	116.2	0.53 (0.52–0.55)
Total	116,022	17,194	2,304,026	41,146	23.9	178.6	−

## Data Availability

The data supporting the findings of this study are available from the corresponding author upon reasonable request.

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
