# Peer review of "Differences in Immunological Evasion of the Delta (B.1.617.2) and Omicron (B.1.1.529) SARS-CoV-2 Variants: A Retrospective Study on the Veneto Region’s Population"

_ijerph, 2022, doi:10.3390/ijerph19138179_

Round 1

Reviewer 1 Report

1. Introduction: It worth mentioning that on one hand, the efficacy of the existing vaccines was found to be reduced against the Omicron variant (the authors did mentioned it, but it should base on more references), and on the other hand, there are some important works which showed that the existing vaccines were efficient in reducing the severity and the fatality of the disease among infected vaccinated people. I think that these works should be mentioned in this manuscript.

2. Methods: the authors defined "the Delta period" and "the Omicron period". How these periods were defined? Is there an overlap between these periods in the Veneto county?

Author Response

Cover letter in response to the reviewers’ comments

We are very grateful to the reviewers which comments have certainly helped us to both improve the quality and notice some other limitations of our study. We tried to carry out all the changes suggested by the reviewers, and in the present cover letter we also concisely mentioned how the changes have been made.
In the few cases in which we were not able (or we preferred to not accomplish) suggested modifications, we tried to extensively motivate the reasons of our rebuttal.
The manuscript has been entirely modified in the revision mode of Microsoft Word, so that the reviewers should be able to see all the changes from the previous version. Along with major revisions suggested by the reviewers, some typos or grammatical errors have been corrected.
Explanations of our changes are here presented as point per point replies to reviewers’ comments and suggestions

#1 (First referee)

  1. Introduction: It worth mentioning that on one hand, the efficacy of the existing vaccines was found to be reduced against the Omicron variant (the authors did mentioned it, but it should base on more references), and on the other hand, there are some important works which showed that the existing vaccines were efficient in reducing the severity and the fatality of the disease among infected vaccinated people. I think that these works should be mentioned in this manuscript.
  2. We appreciate the suggestion offered by the reviewer. For the revised version of the article, we added two more references about the reduction in the protection against Omicron’s infections afforded by anti-COVID-19 vaccines ([16-17]) and we also mentioned two other studies showing that, on the other hand, anti-COVID-19 vaccines are still efficient in reducing the severity and the fatality of the disease among vaccinated people infected by the Omicron variant ([19-20]).
  3. Methods: the authors defined "the Delta period" and "the Omicron period". How these periods were defined? Is there an overlap between these periods in the Veneto county?
  4. We apologize to the reviewer if we have not been clear enough about this crucial point. As we wrote in the Material and Methods section, those periods have been defined as follows: Delta period from 1 to 25 November, Omicron period from 1 to 25 January.
    We used these definitions basically because regional databases available at the time of writing were updated to 25 January, and we wanted to build two equal-length time periods characterized by a significant predominance of, respectively, Delta and Omicron variant.
    According to IZSVE (Istituto Zooprofilattico delle Tre Venezie), which is an organization established by national and regional laws as a technical and scientific instrument of the National Health Service, there was an almost absolute predominance of Delta variant up until at least early December 2021, when the first Omicron case on the Veneto Region was sampled. On the other hand, Delta and Omicron variants might have been overlapped to a certain extend during what we defined Omicron period. However, IZSVE estimated that on 3rd January 2022 the prevalence of Omicron variant in SARS-CoV-2 positive samples in the Veneto Region was around 66%, and this percentage kept rapidly increasing during the same month [26]. So, it is safe to say that on January 2022 Omicron variant was predominant.
    Nonetheless, we are aware that this overlap is a limitation of our study and we diffusively mentioned it in the discussion.

Reviewer 2 Report

The unprecedented impact of COVID19 on global health and finance lead to revolutionary development and application of new vaccines. However, SARS-CoV-2 continued to evolve, with new VOCs rapidly transmitting and in parallel, raising concerns over the waning immunity of both previous infections and vaccination. The Veneto region located in the North-East of Italy was one of the most affected in that part of Europe by rising infection rate in both Delta and Omicron SARS-CoV-2 infection waves, along with the increasing vaccine coverage. This retrospective study is based on data analysis from a regional databases compiling demographic and data on COVID19 testing and vaccination. The most important findings of this study are focused on the immunological evasion of Delta and Omicron VOCs in individuals vaccinated with 3rd dose and the ones recovered from the previous infection vs. immunologically naive subjects. Data suggest a diminished protection from Omicron compared to Delta VOC in both previously infected and booster dose vaccinated individuals, focusing on immune evasion rather than on transmissibility of Omicron in its impact analysis. Interestingly, previously infected ('naturally immunized') subjects were the ones with the most significant proportional decrease in their protection from reinfection during the Omicron period. Also, unlike with the natural immunity for the Delta variant which wanes slightly over time, the timing of previous SARS-CoV-2 infection seems to loose its importance during the Omicron period. 

Overall, I find this study to be very important and its findings interesting and highly significant not only for Italy but also for SARS-CoV-2-related public health policies planning throughout the Europe and even globally, emphasising that the vaccine booster is more effective and more reliable than natural immunisation (previous infection) when it comes to protection from SARS-CoV-2 infection, even the Omicron strain.

Author Response

Cover letter in response to the reviewers’ comments

We are very grateful to the reviewers which comments have certainly helped us to both improve the quality and notice some other limitations of our study. We tried to carry out all the changes suggested by the reviewers, and in the present cover letter we also concisely mentioned how the changes have been made.
In the few cases in which we were not able (or we preferred to not accomplish) suggested modifications, we tried to extensively motivate the reasons of our rebuttal.
The manuscript has been entirely modified in the revision mode of Microsoft Word, so that the reviewers should be able to see all the changes from the previous version. Along with major revisions suggested by the reviewers, some typos or grammatical errors have been corrected.
Explanations of our changes are here presented as point per point replies to reviewers’ comments and suggestions

#2 (Second Referee)

The unprecedented impact of COVID19 on global health and finance lead to revolutionary development and application of new vaccines. However, SARS-CoV-2 continued to evolve, with new VOCs rapidly transmitting and in parallel, raising concerns over the waning immunity of both previous infections and vaccination. The Veneto region located in the North-East of Italy was one of the most affected in that part of Europe by rising infection rate in both Delta and Omicron SARS-CoV-2 infection waves, along with the increasing vaccine coverage. This retrospective study is based on data analysis from a regional databases compiling demographic and data on COVID19 testing and vaccination. The most important findings of this study are focused on the immunological evasion of Delta and Omicron VOCs in individuals vaccinated with 3rd dose and the ones recovered from the previous infection vs. immunologically naive subjects. Data suggest a diminished protection from Omicron compared to Delta VOC in both previously infected and booster dose vaccinated individuals, focusing on immune evasion rather than on transmissibility of Omicron in its impact analysis. Interestingly, previously infected ('naturally immunized') subjects were the ones with the most significant proportional decrease in their protection from reinfection during the Omicron period. Also, unlike with the natural immunity for the Delta variant which wanes slightly over time, the timing of previous SARS-CoV-2 infection seems to loose its importance during the Omicron period. 

Overall, I find this study to be very important and its findings interesting and highly significant not only for Italy but also for SARS-CoV-2-related public health policies planning throughout the Europe and even globally, emphasising that the vaccine booster is more effective and more reliable than natural immunisation (previous infection) when it comes to protection from SARS-CoV-2 infection, even the Omicron strain.

Response

We really appreciate the comment of the reviewer. We are aware that our study is not without limitations, which have all been discussed through the discussion. But, we still believe that our findings could be highly significant for SARS-CoV-2 related public health policies, as the reviewer wrote.
 We also agree with the reviewer when he/she said that these results could be important globally and not just in Italy, because of that we think that the design of our study makes the analysis very generalizable, since it considers a broad class of people without focusing on specific subgroups for whose results may be not extendable to different populations.

Reviewer 3 Report

See attached

Author Response

Cover letter in response to the reviewers’ comments

We are very grateful to the reviewers which comments have certainly helped us to both improve the quality and notice some other limitations of our study. We tried to carry out all the changes suggested by the reviewers, and in the present cover letter we also concisely mentioned how the changes have been made.
In the few cases in which we were not able (or we preferred to not accomplish) suggested modifications, we tried to extensively motivate the reasons of our rebuttal.
The manuscript has been entirely modified in the revision mode of Microsoft Word, so that the reviewers should be able to see all the changes from the previous version. Along with major revisions suggested by the reviewers, some typos or grammatical errors have been corrected.
Explanations of our changes are here presented as point per point replies to reviewers’ comments and suggestions

#3

Comments Cocchio et al. have performed a retrospective analysis using data retrieved from regional surveillance databases to evaluate the protection against SARS-CoV-2 infection offered by natural immunity and a three-dose vaccination regimen. The analysis was performed separately (stratified) on data from two distinct time periods representative of the Delta and Omicron variant. Despite the limitations, the study findings support the fact that completing a three-dose vaccination cycle provides the highest protection against both variants.

Point 1: Abstract, L16: define “recently”.

  1. We apologize to the reviewer for the ambiguous term that we have now corrected, specifying that the unprecedent wave of infection faced by the Veneto Region occurred in December 2021 – January 2022.

Point 2: General comment: Please be consistent when referring to the variants throughout the manuscript. Either use “Delta/Omicron”, or use “B.1.617.2/B.1.1.529” (e.g., “Omicron” in L27 and “B.1.1.529” in L28).

  1. We appreciate the reviewer’s suggestion. For the sake of both brevity and clarity, after introducing them also with the scientific nomenclature, in the revised version of the manuscript we now always use the terms Delta and Omicron.

Point 3: General comment: As explained in L136-139, there is no distinction between individuals who received an additional dose or a booster dose. To avoid confusion between different vaccination schedules, I would suggest to refer to this group as “three-dose regimen” instead of referring to a “booster dose”. Again, please be consistent throughout the manuscript (e.g., “third dose” in L27 vs. “booster vaccination cycle” in L31). 

  1. We appreciate the reviewer’s suggestion. In the new version of the manuscript, after explaining the different concept of additional and booster dose, we always refer to the group as three-dose regimen, as suggested by the reviewer. Also labels in the two figures were now changed according to this new notation.

Point 4: Introduction, L53: Rephrase “extended to all the population”.

  1. We apologize to the reviewer for the lack of clarity. We now added the entire time-evolution of the vaccination schedule, according to national guidance. We also added previously missing information about booster doses, which is the fact that they are available just for subjects above the age of 12.

 Point 5: Introduction, L71-73: If available, please add the coverage of the booster doses, similarly as provided in L56-57.

  1. Unfortunately, the source we used for the coverage of first doses and regular primary cycles – which is the official web-portal of vaccination of the Veneto Region [6] – does not provide the coverage of booster doses. However, we agree with the reviewer that this data – updated to the time of first writing of the article, which was approximatively the beginning of March 2022 - could be important.
    Therefore, we tried to search this information online, and the most reliable source we found is a local journal which on 8th March 2022 reposted the daily monitoring of the anti-COVID-19 campaign directly from the regional council of the Veneto Region. According to that source, coverage of third doses was equal to 64.6% [11].
    Since it is plausible that small deviations between this source and the one we used for the coverage of primary cycle occur, to be extremely cautious we mentioned that – at the time of writing – the coverage for booster doses was around 65%.

Point 6: Introduction, L76-82: This is a complex sentence. Please rewrite.

  1. We apologize to the reviewer for the lack of clarity. We tried to rewrite these sentences in a more comprehensible way.

Point 7: Introduction, L94: Consider rewriting this sentence, especially the part “to the disadvantage of the previous Delta one on …”

  1. We tried to rewrite this sentence in a more concise and comprehensible way, without changing its meaning.

 Point 8: Methods, L106: Specify which test results were included (e.g., RT-PCR tests, rapid antigen tests).

  1. We apologize to the reviewer if the fact that we are considering just RT-PCR tests was not that clear. We now explicitly wrote through the Material and Methods section that we excluded rapid antigen tests from the analysis. We did not mention the reasons of this inclusion criteria since they are already extensively discussed through the Discussion, as this exclusion may be considered as a limitation of the present study.

Point 9: Methods, L108-109: A suspected reinfection was defined as a positive PCR test for SARS-CoV-2 ≥90 days after the first positive test in accordance with CDC investigative reinfection criteria (Yahav et al.). However, the minimum interval between episodes described in the definition of a reinfection should be reconsidered when a variant with distinct antigenic properties emerges.  To date, reinfections with SARS-CoV-2 are defined by the European Centre for Disease Prevention and Control (ECDC) as two positive tests ≥60 days apart (technical report April 8, 2021). As such, especially for the Omicron time period, using the definition of ≥90 days might underestimate the true number of reinfections. As the natural immunity group has been stratified on the timing of the first infection, I suggest to also include those with a previous infection ≥60 days before the beginning of the study. 

  1. We are aware that on 8th April 2021 ECDC proposed the case definition for reinfection mentioned by the reviewer. However, it has also to be considered that currently there is not a globally accepted case definition.
    Many countries and public health entities, in fact, still follow their specific case definition for reinfection, and for many of them (f.e. CDC [https://www.cdc.gov/coronavirus/2019-ncov/hcp/clinical-care/clinical-considerations-reinfection.html], UK Health Agency [https://ukhsa.blog.gov.uk/2022/02/04/changing-the-covid-19-case-definition/] etc.) it still coincides with the “>90 days criteria” adopted in our manuscript. Moreover, also when looking at the recent published scientific literature on SARS-CoV-2 reinfection, it turns out that there is no a univocal case definition for reinfection, even though the “>90 days criteria” still seems to be one of the most common ([20], [https://www.nejm.org/doi/full/10.1056/NEJMoa2118946] etc.).
    We agree with the reviewer that the “60 days threshold” could be helpful to avoid the underestimation of the reinfections in correspondence of the rise of a new variant with distinct antigenic properties. However, we believe that - on the contrary - the same threshold could lead to an overestimation of reinfections if applied in a period characterized by the predominance of a single variant. Our study aims to compare protection levels afforded by a preceding infection under the same timing w.r.t. the baseline, and in two different periods each characterized by a single variant. Since we are not describing the burden of reinfections as a whole over time, we think it is preferable to use a conservative/commonly accepted case definition, yet adaptable to both the considered periods.
    All that said, when introducing our choice for the case definition of reinfections in the Material and Methods section we now specified that it is just one possible choice, and that different choices could obviously lead to different results.

Point 10: Methods: Please add the stratification according to the timing of the previous infection.

  1. We appreciate the reviewer’s suggestion. We now added the stratification according to the timing of the previous infection.

Point 11: Methods, L155: Please specify whether all reinfections we considered (irrespective of the timing) for the comparison between the natural immunity and the reference dose.

  1. We appreciate the reviewer’s suggestion. We have now tried to be clearer about the way in which we firstly estimated the protection afforded by a preceding infection with a precise timing, and then also a global protection afforded by a preceding infection irrespective of its timing.

Point 12: Methods, L134: Related to Point 3, please state more explicitly whether for example immunocompromised patients who received both an additional dose after their primary vaccination cycle and a booster dose (i.e., a total of four doses) are excluded.

  1. We appreciate the reviewer’s suggestion. However, as described in the official communication from the Italian Ministry of Health (Circolare 0013209 del Ministero della Salute, 20 Febbraio 2022), in Italy the administration of fourth doses for immunocompromised subjects was approved just on 20th February 2022. This means that in the periods considered for our study, there were no Veneto Region inhabitants who received more than four doses. Thus, we think that specifying such an exclusion could lead to a miss-interpretation of the Italian Public Health Policy at the time in which our analysis took place.
    Nevertheless, we now specified that before our analysis vaccination data were cleaned by the presence of possible mistake during data entry. Particularly, subjects included in the study are those with a vaccination schedule consistent with rules and timing expected by Italian Ministry of Health and approved by EMA.

 Point 13: Methods, study design: Would it be an option to create a fourth study population group with hybrid immunity (as suggested in the discussion L297-299)? If not, explain in the discussion why this was not evaluated in the current study.

  1. We particularly appreciate the reviewer’s suggestion. In fact, our first idea for the present work was to also consider this population group: however, probably due to the brevity of the considered periods and the low numerosity of (re)infections among such a subgroup in the considered time intervals, preliminary analysis gave unrobust and poorly statistically significant results about the protection afforded by hybrid immunity. We thus decided to restrict and deepen our analysis just to the three populations currently described in the article, for which results are more robust and reliable. All that said, we now explained this through the discussion and for sure we will personally furtherly investigate that point when more updated data will be available.

 Point 14: Methods, L166: Please be consistent in terminology and replace by “multivariable”.

  1. We appreciate the reviewer’s suggestion. In the revised version of the manuscript, we always adopt the term “multivariable”.

Point 15: Results, L189-190: Related to Point 11, please specify whether the “85% protection given by natural immunity” takes into consideration all reinfection irrespective of the timing. For clarity, it might help to add an extra line in Table 1 for the overall effect of previous infection (irrespective of the timing) to show how this 1-- adjOR was obtained. Same applies to L217 and Table 2.

  1. We appreciate the reviewer’s suggestion. For the revised version of the manuscript, we specified how the global protection afforded by a previous infection was computed and we added an extra line for both the Table.

Point 16 : Results, Table 1 and Table 2: 218 Please specify the period (Delta or Omicron) in the title/legend of the tables.

  1. We appreciate the reviewer’s suggestion. We now specified the considered period in the title of the tables.

 Point 17 : Discussion, L235240: This paragraph describes the changes in incidence rates between the two time periods. However, I doubt whether it is appropriate to state that the observed increase is “due to the rise of Omicron variant”. The results are obtained aft conducted separately on datasets from different time periods. er analyses When comparing the protection from immunity against infection between variants that circulate during distinct time periods, it is important to take into account different potential c onfounders that might also fluctuate over time. Here, potentially not all existing confounders have been taken into account which may hamper the comparison. Therefore, I suggest to only describe the findings from both analyses (conducted in the two time pe riods) without inferring a causeandeffect relationship. Please also add to the limitations that potentially not all confounding variables have been taken into account im . There could exist other underlying factors explaining the change in protection from munity between the two time periods besides the circulating variant characteristic , s of the affected population and healthcare organizational aspects.

  1. We appreciate the reviewer’s suggestion. We are aware that our study is not out of limitations, and our intentions was not to infer a cause-effect relation. Conversely, when we started writing the first version of the manuscript, our intention was to bring to the attention a possible immunological evasion – potentially involving both natural and vaccine-induced immunity - of the Omicron variant which had recently became the new predominant variant on the Veneto Region. We apologize to the reviewer if some of our words have led to a misinterpretation of our intention. We now tried to reformulate some parts of the discussion, also furtherly emphasising the limitations raised by the reviewer.

    Point 18 : Discussion, L281such as the 285: This is an important limitation. Please elaborate on the pot impact of this on the study results (dilution of the effect?).
  2. We appreciate the reviewer’s suggestion. We now elaborated with more detail the potential impact of both the limitations cited by the reviewer.

 Point 19 : Discussion: I wonder what would be the impact of the effects ential of undocumented previous infections on the comparison and the fact that the rate of underascertainment varies con siderably over time. Please elaborate on this in the discussion section.

  1. We particularly appreciate the point raised by the reviewer, since in our opinion the under-ascertainment of SARS-CoV-2 cases represents an issue affecting the global public health and, potentially, the whole scientific literature on SARS-CoV-2 based just on “documented” infections.
    In the specific case of our study, we think that it is very difficult to infer how undocumented (previous) infections could have affected the results.
    On the one hand, it is plausible that undocumented infections occurred homogeneously in both the populations we considered: if that hypothesis were true, one could argue that the undocumented infections occurred in a group balance the ones occurred in the others, keeping reasonably reliable the results currently exhibited. On the other hand, it is also possible that undocumented infections were not distributed evenly in the three groups. For example, one could suppose that unvaccinated subjects are also less prone to being tested, increasing the chance of having an undocumented infection among them. If that hypothesis were true, in contrast with the former case, one could argue that the protection afforded by both natural and vaccine-induced immunity is currently underestimated, since there would be a more significant hidden fraction of naturally immunized subjects in the reference group.
    However, we do not believe to the presence of trustable evidence in support of any assumption of this kind. For instance, returning to the example above, even if it were true that unvaccinated subjects are also less prone to being tested, it should be considered that in Italy for several months they were obliged to obtain green pass (i.e, to be tested negative for SARS-CoV-2) to participate to almost all social activities in public spaces, possibly decreasing the chance of having undocumented infections among unvaccinated during this period.
    All that said, we now mentioned this point during the discussion as we think it represents a further reason why our results should be interpreted carefully.

Point 20 : Discussion: Please clarify whether the died or were lost to followobservations were censored when the subject up. If not, please add this to the limitations.

  1. We appreciate the reviewer’s suggestion. We now added this point when discussing the limitations of our study.

Round 2

Reviewer 3 Report

See attached

Author Response

We agree with all the three further suggestions raised by the reviewer.

Point 8: we now just mentioned the exclusion of rapid antigen swabs in the Material and Methods section, whereas we explained the reasons behind this latter exclusion through the Discussion.

Point 9: we moved the explanation in L121-L128 of the previous version of the manuscript in the Discussion.

Point 15: we corrected a refusal affecting Table 1 and Table 2 kindly noticed by the reviewer.

We are grateful to the reviewer for this further suggestions.